# A Remote Field Course Implementing High-Resolution Topography Acquisition with Geomorphic Applications

Sharon Bywater-Reyes[1], Beth Pratt-Sitaula[2]

[1]Department of Earth and Atmospheric Sciences, University of Northern Colorado, Greeley, Colorado, 80639, United States

[2]Education and Community Engagement, UNAVCO, Boulder, Colorado, 80301, United States

*Correspondence to*: Sharon Bywater-Reyes (sharon.bywaterreyes@unco.edu)

## Abstract

Here we describe the curriculum and outcomes from a data-intensive geomorphic analysis course: "Geoscience Field Issues Using High-Resolution Topography to Understand Earth Surface Processes," which pivoted to virtual in 2020 due to the COVID-19 pandemic. The curriculum covers technologies for manual and remotely sensed topographic data methods, including: 1) Global Positioning Systems and Global Navigation Satellite System (GPS/GNSS) surveys, 2) structure from motion (SfM) photogrammetry, 3) and ground-based (terrestrial laser scanning; TLS) and airborne lidar. Course content focuses on earth-surface process applications, but could be adapted for other geoscience disciplines. Many other field courses were cancelled in summer 2020, so this course served a broad range of undergraduate and graduate students in need of a field course as part of degree or research requirements. Resulting curricular materials are available freely within the National Association of Geoscience Teachers' (NAGT's) *Teaching with Online Field Experiences* collection. The authors pre-collected GNSS data, uncrewed-aerial-system- (UAS-) derived photographs, and ground-based lidar, which students then used in course assignments. The course was run over a two-week period and had synchronous and asynchronous components. Students created SfM models that incorporated post-processed GNSS ground control points and created derivative SfM and TLS products including classified point clouds and digital elevation models (DEMs). Students were successfully able to 1) evaluate the appropriateness of a given survey/data approach given site conditions; 2) assess pros and cons of different data collection and postprocessing methods in light of field and time constraints and limitations of each; 3) conduct error and geomorphic change analysis; and 4) propose or implement a protocol to answer a geomorphic question.  Overall, our analysis indicates the course had a successful implementation that met student needs as well as course-specific and NAGT learning outcomes, with 91 % of students receiving an A, B, or C grade. Unexpected outcomes of the course included student self-reflection and redirection and classmate support through a daily reflection and discussion post. Challenges included long hours in front of a computer, computing limitations, and burnout because of the condensed nature of the course. Recommended implementation improvements include spreading the course out over a longer period of time or adopting only part of the course and providing appropriate computers and technical assistance. This manuscript and published curricular materials should serve as an implementation and assessment guide for the geoscience community to use in virtual or in-person high-resolution topographic data courses that can be adapted for individual labs or for an entire field or data course.

## 1 Introduction

### 1.1 Background on course format and partners

The COVID-19 pandemic forced most higher education courses to use virtual delivery modes for part or all of 2020 (Ali, 2020), which posed a challenge for all disciplines. This change was particularly challenging for the many United States (US) undergraduate geoscience programs, which require field camp or a field course for degree completion (Wilson, 2016). The majority of these field courses had been planned for in-person implementation and were quickly redesigned for remote delivery. Most US universities closed campuses March 2020 and did not return to in-person until fall 2020 or later; whereas the field courses needed to occur May through August 2020. In response to this crisis, geoscience field instructors worked together with the National Association of Geoscience Teachers (NAGT) to develop and share remote field teaching resources through the *Designing Remote Field Experiences* project (Egger et al., 2021).

This manuscript describes one such impacted course that pivoted to remote teaching, "Using High-Resolution Topography to Understand Earth Surface Processes," taught through the University of Northern Colorado (UNC). It was originally planned as an in-person course with Structure from Motion photogrammetry (SfM), Terrestrial Laser Scanning (TLS), and Global Navigation Satellite System (GNSS)[1] data collection and analysis applied to geomorphic issues in a mixed field/classroom setting. The course implementation and curriculum were adjusted to a remote delivery mode by collecting TLS, GNSS, and uncrewed aerial system (UAS) imagery for SfM prior to the course start. Informational videos about the field site and data collection were also provided to the students. The data were collected near Greeley, Colorado on the Cache la Poudre River by Bywater-Reyes from the University of Northern Colorado, in collaboration with UNAVCO (unavco.org). Other geomorphic datasets were drawn from UNAVCO and OpenTopography (https://opentopography.org/) archives. The class had 23 students total (16 undergraduates and 7 graduate students).

Bywater-Reyes was the primary course designer and instructor for the course and led the adjustments to remote teaching. UNAVCO runs the National Science Foundation (NSF) and National Aeronautics and Space Administration (NASA) geodetic facility (GAGE: Geodetic Facility for the Advancement of Geoscience). Its mission includes providing educational support to the broader geodesy and geoscience communities; thus, UNAVCO staff collaborated on the prepared data collection. The teaching activities developed for this course were adapted from UNAVCO's GEodesy Tools for Societal Issues (GETSI; https://serc.carleton.edu/getsi/index.html) modules: Analyzing High Resolution Topography with TLS and SfM (https://serc.carleton.edu/getsi/teaching_materials/high-rez-topo/index.html) and High Precision Positioning with Static and Kinematic GPS/GNSS (https://serc.carleton.edu/getsi/teaching_materials/high-precision/index.html)

This course, and the activities it included, were contributed to the NAGT *Designing Remote Field Experiences* collection (https://serc.carleton.edu/NAGTWorkshops/online_field/index.html) (Egger et al., 2021). The overall course is at

---

[1] GNSS (Global Navigation Satellite System) is the general term that refers to all Earth's satellite navigation systems. Most people are more familiar with the term GPS (Global Positioning System), which technically, refers to only the US satellite constellation. Hereafter this paper will refer to GNSS or GPS/GNSS.

https://serc.carleton.edu/NAGTWorkshops/online_field/courses/240348.html and the individual activities are linked within
the course page, as well as being contributed individually to the *Online Field Teaching Activities* collection
(https://serc.carleton.edu/NAGTWorkshops/online_field/index.html).

**1.2 Value of course topic**

High-resolution topographic datasets (SfM and ground-based and airborne lidar) are valuable in disciplines ranging from
geomorphology and tectonics to engineering and construction (Bemis et al., 2014; Passalacqua et al., 2015; Robinson et al.,
2017; Tarolli, 2014; Westoby et al., 2012). Use of high-resolution data in Earth Science education allows students to quantify
landscapes and their change at sub-meter resolution (Pratt-Sitaula et al., 2017; Robinson et al., 2017). Understanding surface
processes is listed as very important in the recent "Vision of Change in the Geosciences" with the objective "Students will be
able to recognize key surface processes and their connection to geological features and possible natural and man-made hazards"
(Mosher et al., 2021, p. 17). Furthermore, use of multiple types of data allows students to practise critical thinking skills such
as assessing which acquisition method is appropriate for different scenarios and what errors are associated with different
methods. Critical thinking, integrating diverse data sources, and strong quantitative skills were all identified as very important
skills for undergraduate students to master (e.g., Kober, 2015). Similarly, making inferences about the Earth system; making
spatial and temporal interpretations; working with uncertainty; and developing field, GIS, computational, and data skills were
all listed as very important skills for geoscience students to demonstrate (Mosher et al., 2021). Furthermore, learning to collect,
post-process, and analyse large datasets is a marketable transferable skill that prepares students for the job market, with
cartography and photogrammetry job prospects being "excellent" according to the Bureau of Labor Statistics. For historically
marginalized students, high-paying job prospects are particularly important (O'Connell and Holmes, 2011).

**1.3 Value of remote learning to removing barriers**

Fieldwork, while valuable to building students' self-efficacy and problem-solving skills (Elkins and Elkins, 2007), can pose a
barrier to diversifying the geosciences because of ableism (Carabajal and Atchison, 2020), cost (Abeyta et al., 2020), cultural
factors (Hughes, 2015), racism (Abbott, 2006), and sexism (Fairchild et al., 2021) in the field. COVID-19 forced the
geosciences to develop virtual field experiences, with a positive side effect of removing many of the aforementioned barriers
to fieldwork completion. For example, the computer-based nature of remote field learning removes many physical accessibility
issues present for typical field courses. The option to learn from home may make the remote courses more feasible for students
with family or work responsibilities, as well as reduce real and perceived safety issues related to gender, sexual orientation,
and race that may occur in tradition field camp settings. Although remote field courses are not necessarily the most desirable
for all students, the development of high quality remote field options can be one component of diversifying the geosciences
(Egger et al., 2021).

## 2 Course overview and learning outcomes

### 2.1 Course objectives and geodetic methods

The objective of the course was for students to learn manual and remote sensing methods of topographic data collection, including 1) GPS/GNSS, 2) SfM, and 3) TLS surveying and airborne lidar use. GNSS uses ground-based receivers to trilaterate positions calculated from signals sent by orbiting satellites (to accuracies of a couple centimetres in this use case). SfM is a photogrammetric technique that uses overlapping images to construct three-dimensional models with widespread research applications in geodesy, geomorphology, structural geology, and other subfields in the geosciences (Passalacqua et al., 2015; Westoby et al., 2012). Lidar also generates three-dimensional models valuable for the same range of applications; but it uses laser scanners to send out thousands of laser pulses per second, measure the return time, and calculate distances. Scanners can be ground-based (TLS) or airborne. SfM requires less expensive equipment and less field time, but more processing time, than TLS. In low-vegetation field areas, SfM can yield similarly valuable high-resolution topographic models with point densities usually hundreds of points per square metre (depending on instrument-to-object distance; Westoby et al., 2012); however, TLS is much more effective in areas with dense vegetation. For both methods, ground control points (GCP), usually measured with GNSS, are needed for georeferencing the topographic model. For SfM, they are also critical for reducing distortions and errors (James et al., 2019). One of the key outcomes for students was to understand the benefits and challenges of each method and how to determine the most valuable in different circumstances.

### 2.2 Course delivery

Course content focused on earth-surface process applications, but could be adapted to other geoscience topics. The course was taught workshop style, composed of multiple synchronous work sessions with asynchronous work time in between. The bulk of the instruction occurred within a 2-week period during the summer. Synchronous lectures were conducted via Zoom and course content distributed via Canvas. The class used Slack as an asynchronous way to exchange questions, comments, and solutions amongst the students and between the students and instructor. During the course, students worked with three different analytical software packages: Agisoft MetaShape, CloudCompare, and ArcGIS Map. Five students attended an optional in-person field collection campaign (one student travelled from out-of-state and the remainder were UNC students). The course was divided into two units: Unit 1 focused on the SfM workflow, including integrating GNSS and point cloud processing; and Unit 2 on lidar products and workflows, including TLS, topographic differencing, airborne lidar, and methods comparison. Each unit ended in a unit report, with the second providing students an opportunity to improve workflows and explore additional data sources and analyses.

### 2.3 Learning outcomes

The course-specific learning outcomes were, students should be able to:

| 126 | | A. | Make necessary calculations to determine the optimal survey parameters and survey design based on site and available |
| 127 | | | time. |
| 128 | | B. | Integrate GNSS targets with ground-based lidar and SfM workflows to conduct a geodetic survey. |
| 129 | | C. | Process raw point cloud data and transform a point cloud into a digital elevation model (DEM). |
| 130 | | D. | Conduct an appropriate geomorphic analysis, such as geomorphic change detection. |
| 131 | | E. | Justify which survey tools and techniques are most appropriate for a scientific question. |

132

The course activities also helped students meet many of the NAGT Capstone Field Experience Learning Outcomes. These nine outcomes were developed by a group of 32 experienced field educators, who came together in spring 2020 to develop comprehensive learning outcomes for field experiences that are relevant to both in-person or online delivery modes (https://serc.carleton.edu/NAGTWorkshops/online_field/learning_outcomes.html). By the end of a capstone field experience, whether that experience is online or in-person, students should be able to:

1. Design a field strategy to collect or select data in order to answer a geologic question.
2. Collect accurate and sufficient data on field relationships and record these using disciplinary conventions (field notes, map symbols, etc.).
3. Synthesize geologic data and integrate with core concepts and skills into a cohesive spatial and temporal scientific interpretation.
4. Interpret earth systems and past/current/future processes using multiple lines of spatially distributed evidence.
5. Develop an argument that is consistent with available evidence and uncertainty.
6. Communicate clearly using written, verbal, and/or visual media (e.g., maps, cross-sections, reports) with discipline-specific terminology appropriate to your audience.
7. Work effectively independently and collaboratively (e.g., commitment, reliability, leadership, open for advice, channels of communication, supportive, inclusive).
8. Reflect on personal strengths and challenges (e.g. in study design, safety, time management, independent and collaborative work).
9. Demonstrate behaviors expected of professional geoscientists (e.g., time management, work preparation, collegiality, health and safety, ethics).

Table 1 shows the alignment between the daily activities and course-specific and NAGT learning outcomes. It also provides links to the activity pages within the NAGT *Online Field Teaching Activities* collection.

**2.4 Field site and prepared data**

The course field site was the Cache la Poudre River at Sheep Draw Open Space (City of Greeley Natural Areas) in northern Colorado. It was selected because: 1) the site shows both standard river features and evidence of extreme flooding; 2) the

Poudre River is important to several local communities; and 3) the site is proximal to the UNC campus. According to the
Coalition for the Poudre River Watershed, "The Cache la Poudre River Watershed drains approximately 1,056 square miles
above the canyon mouth west of Fort Collins, Colorado. The watershed supports the Front Range cities of Fort Collins, Greeley,
Timnath and Windsor. In an average year, the watershed produces approximately 274,000 acre feet of water. More than 80
percent of the production occurs during the peak snowmelt months of April through July"
(https://www.poudrewatershed.org/cache-la-poudre-watershed). In 2013, the Front Range and plains of Colorado experienced
extensive flooding. The region received the average annual rainfall in one week (Gochis et al., 2015). There was extensive
damage to infrastructure and in some cases the erosion of a 1000-years' worth of weathered material (Anderson et al., 2015).
Near Greeley, significant portions of the Poudre trail were impacted as the river topped its floodplain and eroded its banks.
The study site is adjacent to the Poudre Trail, with portions of the former trail eroded into the river and the current trail rerouted
around the 2013-developed river course.
Data for student use was collected from the Poudre River by a joint UNAVCO-UNC team in May 2020. The types of data
included were:
- UAS-collected photographs for SfM point cloud generation (DJI Mavic 2 Pro)
- Point clouds collected using TLS (Riegl VZ400)
- Several hours of GNSS base station data (Septentrio Altus APS3G)
- GNSS-measured ground control points locations for georeferencing both SfM and TLS surveys (Septentrio Altus
APS3G)
- Videos of field site and field methods
**3 Methods**
This course was developed and implemented in response to the COVID-19 pandemic and the need for students to fulfil degree
requirements and not designed as an educational research study before implementation. Thus, there are inherent limitations to
the available data and conclusions that can be drawn from the project. Nonetheless, there is value in sharing this robust open-
source curriculum, describing how the course was implemented, and outlining how student learning outcomes were assessed
and achieved. This study went through the Institutional Review Board at University of Northern Colorado, which determined
this project to be exempt under 45 CFR 46.104(d)(704) for research, Category 4. Therefore, course artefacts and student
demographic data can be used in research so long as no identifying information is revealed. Student artefacts included
submitted assignments, unit reports, posts from a daily Slack discussion forum and unsolicited feedback given directly to the
instructor. We extracted examples from artefacts and associated assessments to illustrate students' accomplishments and
evaluate whether the course, and to a lesser extent, NAGT capstone field learning outcomes were met. We describe Course
Implementation and Assessment Approach in Section 4, and alignment with course-specific (5.1) and other outcomes (5.2) in
Section 5. We finish with Lessons Learned and Implementation Recommendations in Section 6.
**4 Course Implementation and Assessment Approach**
This section gives a brief overview of each course activity (Table 1) and which Course-specific Learning Outcomes and NAGT
Outcomes are at least partially addressed. Table 2 is an example of the type of rubric used in grading simple student assignment
answers, such as in daily assignments, with discretion used to assign percentages within these ranges. Most questions also had
points-possible indicated so that students could gauge their relative significance towards the grade. Multi-component rubrics
were used for more in-depth exercises, such as unit reports. In such cases, students were informed of the weighted percent for
each section (e.g., title, abstract, introduction, etc.) and also given a detailed description of what should be included in each
(https://d32ogoqmya1dw8.cloudfront.net/files/NAGTWorkshops/online_field/courses/sfm_feasibility_report.v2.docx). The
same simple rubric (Table 2) was used to assess each weighted section. For example, the Discussion section was weighted 20
% and students were instructed:
*"Here, you can discuss both pros and cons of the methods (What worked? Didn't work? What would improve the workflow?)*
*as well as what you discovered about the Poudre River at the site. Return to the question of feasibility. Consider the overall*
*goal of using SfM to assess geomorphic processes on the Poudre River at Sheep Draw. How could SfM be applied? What are*
*the limitations?"*
Similarly detailed instructions accompanied all components for the more in-depth exercises.
**4.1 Day 1: Getting started with Structure from Motion (SfM) photogrammetry**
*Course Unit 1: SfM and GPS/GNSS* started out on Day 1 with an introduction to the SfM method. The day's activities were
the first step in addressing Course Outcomes A (survey design) and C (point cloud data). After an overview presentation
students used smartphone cameras to take ~20 overlapping photos of an object of interest (ex. sofa, shed, berm). For simplicity
and to learn about local reference frames (rather than global ones from GNSS) they took compass bearing, inclination, and
distance measurements and used trigonometry to calculate X-Y-Z coordinates for the ground control points (GCP). Students
used Agisoft MetaShape software to post-process their photos and create the 3D point clouds. The software was available on
their personal computers through a 30-day trial licence. Students then evaluated the performance of their model by considering
data quality in different model regions and what method changes might improve their product. They also made
recommendations for how SfM could be applied to different fields in the geosciences. The assessment of student learning was
based on successful production of a locally-referenced point cloud and the data quality analysis.

**4.2 Day 2: Introduction to GPS/GNSS**

In the Day 1 activity, students used a relative local coordinate system to produce an accurately-scaled model. However, for real-world applications a global coordinate system is frequently preferable, which can be achieved with survey-grade GPS/GNSS; so Day 2 was focused on Course Outcomes A (survey design), B (GNSS and geodetic survey), and E (justifying techniques). Day 2 morning activities were adapted from the GETSI module *High Precision Positioning with Static and Kinematic GPS/GNSS.* First students learned about the method through a lecture. Next they worked with data collected using different types of receivers and resulting accuracy and precision. Assessment included a concept sketch of GPS/GNSS systems, quantification and evaluation of accuracy and precision of different grades of GNSS, and recommendations for appropriate applications of each.

In the afternoon of Day 2, students were introduced to the field site and methods used for data collection at the Cache la Poudre field location (described above in Section 2). Students watched a video (Video 1; https://youtu.be/EZ5I8Ge8YjI) about the field site and a video introducing the GNSS methods (Video 2; https://youtu.be/Xpj1QJf8AkY). Then, using the pre-collected base-station data, students completed the assignment *Post-Processing GPS/GNSS Base Station Position*. Students submitted the base station file to the Online Positioning User Service (OPUS), the National Geodetic Survey (NGS)–operated system for baseline processing of standardised RINEX files into fixed (static) positions. For the assessment, students wrote a paragraph explaining their procedure, interpreting the results, describing the difference between ellipsoid height and orthometric height, and highlighting anything that was surprising or confusing about the results.

**4.3 Day 3: SfM of Poudre River at Sheep Draw Reach**

On Day 3 students combined skills learned in the previous two day in order to create a georeferenced point cloud from the field site (Course Outcomes A-C) and started to consider relevant geomorphic analyses (Outcome D). The morning exercise was *Ground Control Points and SfM at Cache la Poudre Site*. This began with a group discussion on where ground control points at the site should be placed within the field area (Figure 1). Students were then given a text file of the x, y, z coordinates (UTM) collected by the UNAVCO-UNC team, and had to import them into ArcGIS to create a ground control point map. In a follow-up discussion, students compared the ground control point locations actually used in the prepared data with the locations they discussed for placement in the initial discussion. They were asked to summarise the strengths and weaknesses of the implemented ground control point plan at the site, which helped to assess learning related to both survey design outcomes.

The afternoon exercise was *Structure from Motion for Analysis of River Characteristics*. Students picked either Area of Interest 1 or 2 for their SfM workflow (Figure 1). Students with adequate computing power could choose to do the entire study region. Using resolution and height information about the UAS-collect photographs, students first calculated the expected resolution of the final point cloud. They were then asked to assess what types of features or processes at the Cache la Poudre study area they expected could be resolved; from there, they discussed the types of geomorphic questions they could

feasibly expect to answer with the dataset of that resolution. Next students followed a more detailed Agisoft MetaShape Guide
to construct a georeferenced point cloud of their Area of Interest. As they were familiar with MetaShape from Day 1, students
were able to work through the procedure independently. Once their model was complete, students were asked to answer a
series of questions related to error analysis of their model and to reassess appropriate geomorphic applications and design of
the ground control point network used. Finally, students were asked to formulate a testable hypothesis related to processes on
the Cache la Poudre River that they could answer with their dataset. For example, students could investigate cutbank stable
bank heights and angles. The completed exercise was the summative assessment and particularly revealed student
accomplishment of SfM point cloud creation and geomorphic analysis.

### 4.4 Day 4: Using CloudCompare and Classifying with CANUPO

On Day 4, students used the open-source software CloudCompare (http://www.danielgm.net/cc/), which allows for viewing
and manipulation of point clouds. This was a continuation of the same learning outcomes as Day 3 afternoon (Outcomes C
and D) and continued on to some justification of methods (Outcome E). Students learned the basic operations used in
CloudCompare, such as importing point clouds, classifying the points, and taking measurements that allow for hypothesis
testing. They also incorporated an open-source plugin called CANUPO (http://nicolas.brodu.net/en/recherche/canupo/) that
facilitates additional point cloud classification (Brodu and Lague, 2012), such as distinguishing between vegetation and
ground. Students create a digital elevation model (DEM) from ground points and export it for use in ESRI ArcGIS Map. In
ArcGIS, students familiarized themselves with viewing 3D data in 2.5D and created hillshade and slope maps. Then they were
asked to retest their hypothesis with tools available in ArcGIS and 2.5D (e.g. measure tool, raster values). Students compared
and contrasted applications with the three-dimensional point cloud versus 2.5D raster and summarised the appropriate uses
and applications of each in the day's assignment.

### 4.5 Day 5: SfM Feasibility Report Assignment

The summative assessment for Course Unit 1 was the *SfM Feasibility Report*, which included assessment of all five Course
Outcomes. Students were to imagine themselves as natural resource managers and assigned the task of investigating the
feasibility of using SfM to study geomorphic processes on the Cache la Poudre River. They were asked to summarize the SfM
workflow and present the outcomes, limitations, and suggested applications of their SfM model of their Poudre Area of Interest.
On Day 5, students were given a work day to complete the report.

### 4.6 Day 6: Optional Field Trip

Day 6 consisted of an optional field demonstration during which students completed a GNSS ground control survey and
Bywater-Reyes and colleagues collected UAS images at the Poudre Learning Center (https://youtu.be/s5CGhk8GIOU;
Bywater-Reyes, Sharon: Poudre Learning Center Project. https://doi.org/10.5446/54388).

**4.7 Day 7: Introduction to Terrestrial Laser Scanning (TLS)**

Day 7 was the start of *Course Unit 2: TLS, Topographic Differencing, and Method Comparison* and began with an introduction to TLS methodology through a video and lecture. The exercise used pre-collected TLS data that the students were asked to compare and contrast with the SfM point cloud they had developed in Unit 1, which was collected from the same geographic location (Cache la Poudre River) on the same day (Figure 3). The learning outcomes primarily focused on Outcome C (point clouds) but also laid the groundwork for more advanced method comparison to come (Outcome E). Students visually inspected the datasets for similarities and differences; then they measured geomorphic features in the scene and compared their measurements for the two methods. Using skills gained in previous class activities, students classified the TLS cloud into vegetation and ground, exported the ground cloud as a text file, and created a raster that matched the specifications of the one made in the SfM activity. This prepared for 3D (cloud-to-cloud differencing) and raster differencing in Day 8. Assessment (mostly formative) was based on their completion of measurements and a discussion of methods comparison, including a group discussion.

**4.8 Day 8: Point Cloud/Raster Differencing and Change Detection**

On the morning of Day 8 students used the concepts of point cloud and raster differencing to further compare their SfM and TLS results and interpret differences between the methods (Outcomes C and E). After a lecture on point cloud differencing, students proceeded with differencing of the SfM and TLS data for their area of interest using CloudCompare with the M3C2 Plugin (Lague et al., 2012). Since these datasets were collected at the same place on the same day, differences between the datasets were due to errors or uncertainties in one or both of the models. Students were asked to interpret the 3D differences between the datasets. The second lecture, on raster differencing, discussed best practices in preparing rasters for differencing (Wheaton et al., 2010). Students then used ArcGIS Raster Calculator tool to subtract one raster from the other. Students interpreted the results and compared the differences between 3D (point cloud), and 2.5D (raster) differencing. The summative assessment was the assignment in which students interpreted their results as an error analysis and discussed which dataset they think is more accurate (and why) and which method provided the most robust error analysis.

So that the students could gain experience with airborne lidar data and with actual geomorphic change detection, during Day 8 afternoon they were given two lidar-derived raster datasets collected before and after the 2013 floods of the Colorado Front Range on a river (South St. Vrain Creek) that experienced substantial geomorphic change. In the exercise *DEM of Difference* students practised raster differencing skills in the context of geomorphic change detection and also characterised their detection limit with a simple thresholding approach. This helped to further address Outcomes C and D as students answered questions in the assignment about the differencing method and made a series of calculations that pertained to geomorphic change.

**4.9 Day 9: OpenTopography Data Sources and Topographic Differencing**

To broaden student knowledge of available data availability, Day 9 focused on additional high resolution (usually lidar) data sources. After a lecture, students conducted an assignment using existing high-resolution datasets housed within OpenTopography (OT; https://opentopography.org/). First, students practised downloading and viewing data from OT; second students conducted a topographic differencing exercise (Crosby et al., 2011), complementing the point cloud and raster differencing students conducted in Day 8. As with Day 8 afternoon, the learning outcomes primarily focused on point clouds and geomorphic analysis (C and D). The learning assessment was done via the student assignment, in which students determine erosion and deposition in a dune field and analyse error and detection thresholds.

**4.10 Days 10 and 11: Methods Comparison Report and Presentation**

The summative assessment for Course Unit 2 and the course as a whole was the final *Methods Comparison Report* and presentations in the last two days of the course. Students picked from a variety of options including: improving methods from Unit 1 (SfM and TLS methods), adding new elements to Unit 1, choosing an additional exploration with the datasets collected on the optional field day, or using a different dataset such as airborne lidar. As the course summative assessment the report pulled together student learning on all five course outcomes. The presentation (Day 11) additionally gave students practice in oral presentation of scientific findings.

# 5 Results

**5.1 Course-specific learning outcomes**

This section provides a variety of examples of how students met the different course-specific learning outcomes. It is not intended to be exhaustive but to provide general illustrations of student learning drawn from both assignments and Slack daily reflections and discussions.

**5.1.1 A) Make necessary calculations to determine the optimal survey parameters and survey design based on site conditions and available time**

In the GNSS/GPS accuracy and precision activity (Day 2), students showed their ability to evaluate appropriate GPS/GNSS techniques in different contexts with the GPS/GNSS error analysis activity (Day 2). Students calculated and compared accuracy and precision of different GNSS/GPS methods and (Day 2) explained which types of surveys or research applications are appropriate for each. Students received an average of an 89 % of this assignment (exemplary), evidence of their ability to link calculations to applications. Students also completed a concept sketch of GNSS systems (Figure 4) describing what factors can interfere with GNSS performance.

338

In the SfM activity (Day 3), students calculated the pixel resolution resulting from the flight parameters used in the pre-collected UAS images and assessed the appropriateness of this resolution to resolve features within the flight. One student wrote in their assignment, "*Obviously the larger scale features will be resolved, like the eroded bank, point bar, and the sidewalk panels in the river, as well as most sizes of vegetation. If the sampling is 0.3-0.5 centimetres per pixel, then it should be able to resolve grasses, and just about any size of gravel. The difference between the water surface and adjacent should be pretty well resolved as well.*" They were also given the UAS flight time for the survey. Thus, students could easily adapt this approach to calculate the time it would take to accomplish a flight reaching the desired resolution for a given application. The discussion of implementation of ground control at the field site (Day 3) allowed students to compare the actual implementation with literature-recommended protocols to discuss strengths and weaknesses given the site conditions (Figure 5). Students also showed the ability to discern improvements to the survey plan given the site condition. For example, one student wrote: "*I think the GCPs* [Ground Control Points] *are very well placed in area-1 and area-2. But the adjoining area of both the areas only got two GCPs- GC4 and GC10 which is too [few]. It may reduce the accuracy of map while joining area-1 and area-2. In addition, area-2 has only one GCP in North direction which may become an issue during georeferencing. To be on safer side we may include one more GCP near GC9 to ensure the coverage of area-2. If only 9 GCPs are available to me then I think the current arrangement of GCP is best.*" Students received an average of 98 % (exemplary) on this discussion, highlighting their ability to evaluate appropriate methods given site conditions.

**5.1.2 B) Integrate GNSS targets with ground-based lidar and SfM workflows to conduct a geodetic survey**

Students used pre-collected GNSS-measured ground control points to georeference the resulting SfM point cloud in the Day 3 SfM activity. As described in the previous section (5.1.1), students integrated the GNSS data into the SfM projects and also discussed the overall survey design and resulting model errors. The suite of activities that used pre-collected GNSS data was successful as indicated by assessment data and student discussions (5.1.1). Whereas students did integrate GNSS targets with an SfM workflow to conduct a geodetic survey, they did not actively integrate GNSS targets for the TLS workflow. The lack of TLS-target integration stemmed from the remote nature of the course and pre-collected nature of the field campaign whereas an in-person implementation would have allowed students to be actively involved with TLS target GNSS data collection and integration. Future remote implementations would need an activity that involves students in TLS GNSS targets data collection and post-processing to meet this learning outcome. However, given the complicated nature of TLS data post-processing, the authors recommend a simple activity such as a discussion of recommended scan locations and a comparison of actual GNSS target locations compared to the recommendation (e.g., similar to that conducted for the SfM field project). In a virtual course format, this learning outcome would need to be edited in the future.

**5.1.3 C) Process raw point cloud data and transform a point cloud into a digital elevation model (DEM)**

Students practised and successfully converted raw point clouds to DEM's several times (Day 4 and Day 7), and also learned how to use the native MetaShape point cloud classification (Day 3) as well as the open-source CANUPO (Day 4) version. When comparing point cloud versus raster elevation products, a student wrote:*"It was hypothesized that SfM methodologies would be best at providing measurements of large-scale elevation changes, however the clear decrease in point cloud density decreased our confidence in these large-scale elevation change measurements along the bank. Small-scale elevation changes along the point bar were best represented by the ArcMap generated hillshade map and DEM while large-scale elevation changes were best represented by the ArcMap generated slope map and DEM. The slope map also had the unique feature of highlighting areas of constant slope and could be used to distinguish between manmade structures and natural vegetation areas in a site of flood damage."* Here, the student showed their ability to recognize pros and cons of point cloud versus raster (DEM) products. Students received an average of 84 % (exemplary) on the raster derivation and manipulation assignment and did even better when they repeated this process. Students received an average of 89 % (exemplary) on the TLS assignment, where they were asked to repeat the process of conducting a quantitative analysis on the cloud, classify the point cloud, extract ground points, and create a DEM, showing their ability to repeat a workflow originally implemented over several days in one step independently to produce a DEM.

**5.1.4 D) Conduct an appropriate geomorphic analysis, such as geomorphic change detection**

With the SfM and TLS field datasets, students recognized the limitation of having only one time snap. A student reported: *"Structure from Motion to assess geomorphic processes on the Poudre River at Sheep Draw is useful and easy to operate. In this project we used SfM to create a model that can measure bank erosion and deposition. However, we did not have enough information to  analyse the rate at which the river was eroding the bank. To conduct this study we would need to conduct several SfM surveys over a length of time to acquire enough variance in data to calculate a rate."* This statement illustrates the student's recognition of the utility of repeat topographic data needed to conduct a geomorphic change analysis that would be appropriate to answer a geomorphic question they had posed.

In the context of comparing SfM and TLS data collected at the field site at the same time, students conducted point cloud and raster differencing (Day 8). Students received an average score of 78 % (low-end of exemplary) on this assignment and extrapolated how one could apply these methods to geomorphic change detection. A student noted in their daily Slack discussion, *"Learning about DoD [DEM of Difference] was a little confusing to me and some of the assignment parts threw me off but other than that I felt like I learned good things today!"* Another that: *"Today's work was a lot more confusing than the last couple days, but it's much more satisfying."* Students illustrated their enthusiasm for manipulated point clouds. A student wrote in their daily discussion, *"Today I enjoyed getting visible products using ArcMap and Cloud Compare."* In comparing the SfM and TLS datasets, a student demonstrated their understanding of how the differencing would be used in the context of geomorphic change by stating: *"During geomorphological analysis, magnitude and*

*direction are both important. Areas that are positive show deposition, while negative areas show erosion."*

401        Students conducted lidar geomorphic change detection with the Day 8 afternoon activity using regional LiDAR

from Colorado 2013 floods and Day 9 (OpenTopography change detection). Students received the lowest assignment scores
on these, with 50 % and 75 %, respectively (basic to minimal performance level). This may indicate a combination of
confusion and burnout two-thirds of the way through the intensive two-week course. 35 % and 17 % of assignments,
respectively, were assigned a 0 % because submissions were missing. If only submitted assignments are considered, average
scores are much higher (76 % and 92 %, respectively), indicating those who were able to stay on top of the dense course
format were able to perform geomorphic change detection to an exemplary level. Students' scores on the Unit 2 report,
which combined elements from the entire course, support the notion that students may have been fatigued and prioritising
assignments worth more points. Average Report 2 scores were the same as Report 1 scores (76 %). One student even went so
far as to download airborne lidar for the Cache la Poudre River and compare SfM, TLS, and airborne lidar for the same area,
showing their ability to combine skill taught in the course and use DEM differencing analysis for either error or geomorphic
change detection, depending on the context.
**5.1.5 E) Justify which survey tools and techniques are most appropriate for a scientific question**
The progression from the introductory SfM project (Day 1) to a field-scale SfM and TLS comparison (Report 2) allowed
students to assess limitations and justify appropriateness of survey techniques to different applications and scientific questions.
Students highlighted where their intro SfM projects (Day 1) produced accurate point clouds and under which conditions the
point clouds had missing data or high error (Figure 6). They were asked to reflect on field applications appropriate for a model
of a similar quality. In the field SfM (Day 3) and TLS (Day 7) activities, students explained where the three dimensional
models had adequate coverage for different applications.

420        For the SfM field assignment (Day 3), students considered model errors (Figure 7) and classification performance in

their assessment of appropriateness for scientific questions. Students received an average of 88 % on the SfM field assignment
(exemplary work), which asked them to think about the questions they set out to answer and discuss whether this would be
possible given the errors and limitations of the model. A student noted:
"*Given the limitations of the model, I'm not sure if I'll be able to answer the question about the vegetation, and I may be able*
*to work on the erosion, but I'm not sure. There are three questions I would like to answer:*
*1.  Can we identify a flood plain in the area?*
*2.  Is the erosion on the bank from normal flow, or the 2013 flooding?*
*3.  Can we determine the erosion rate on the banks?*
*I believe at least the third question can be quantifiable, but the other two might also be quantifiable. The flood plain may be*
*calculated, but a larger image may be needed. The erosion may also be quantifiable. Erosion rate is most likely measurable*
*because we can use the sand bar on the other side of the river as a measure of erosion. Some larger images, and some more*
*up-close images of the bank may be needed to answer these questions.*"

433    Several students observed the limitations of SfM in the presence of vegetation: A student observed: *"Pros of using*

*SFM method is that it can create high resolution data sets at relatively low costs. A negative aspect about this method is that*
*it cannot generate any data through vegetation and so the environment this method can be used in is limited."* Another
student noted: *"Unlike LiDAR technology which is able to image past vegetation and "see" the ground, SfM images cannot*
*see through foliage. While multiple angles of a site can help create ground points beneath vegetation, thick foliage will*
*always have to be removed from the dataset if one is trying to use SfM to create a Digital Elevation Model rather than a*
*Digital Surface Model. Erroneous points below the surface of the water also were prevalent in the 3D point cloud and*
*needed to be removed."* An additional student observation was: *"It would be useful to conduct a study in the summer and*
*winter every year to analyze the change in bank height and distance from the river to the walking path. This method can be*
*done with SfM, but it would be best to use several types of surveying methods to create an accurate set of data because SfM*
*lacks the ability to see beneath trees, vegetation, and the undercut bank due to the drone being 40 to 50 meters in the air.*
*Therefore, terrestrial and airborne lidar should be used to image the areas where SfM lacks."* When comparing SfM and
TLS (Day 7) a student noted in their daily Slack discussion post, "*I was surprised at the difference in quality between the*
*SfM and TLS. I would think TLS would have much higher quality data but perhaps this site was not a prime example of its*
*capabilities.*" These observations show students understood the limitations and appropriateness of SfM and TLS surveying
and also show the ability to improve upon future acquisitions through editing the data collection protocol.

**5.2 Other course outcomes**
**5.2.1 NAGT outcomes**
This course operated under difficult conditions (e.g., global pandemic), but allowed students to meet degree requirements and
accomplish course-specific learning outcomes in addition to meeting many of the capstone field experience student learning
outcomes developed by the field teaching community in collaboration with NAGT (Section 2.3; Table 1). Assessing whether
each NAGT outcome was met is beyond the scope of this manuscript, however, a few that were especially well addressed, and
also those that were not, are highlighted here.

457    NAGT Outcomes 1-5 were practised in many assignments and were highly aligned with course-specific outcomes

(Table 1). Students did not specifically design a field strategy in its entirety (NAGT Outcome 1), but they did assess the
strengths and weaknesses of field strategies and recommend improvements in order to answer a geologic question. This was,
for example, met along with course-specific Outcomes A and B (see Sections 5.1.1 and 5.1.2). They additionally collected
data that allowed them to assess field relationships and record those with both 2D conventional maps (NAGT Outcome 2) as

well as with 3D representations, well-represented by course-specific Outcome C (see Section 5.1.3). A related outcome (NAGT Outcome 6), communicating these products through written products was accomplished through all daily assignments in addition to the two written reports. Verbal communication was accomplished through group discussions as well as group oral presentations at the end of the course, which also aligned with NAGT Outcome 7 (working in a collaborative team). Students synthesised data, integrating information spatially and temporally, to test hypotheses concerning the past/current/future conditions of an earth system using multiple lines of spatially distributed evidence (NAGT Outcomes 3 and 4). In particular, Course-specific Outcome D can be referenced for examples (see Section 5.1.4). Finally, students developed arguments consistent with available evidence and uncertainty (NAGT Outcomes 5) aligns with Course-specific Outcome E (see Section 5.1.5).

The NAGT outcomes that received less intentional attention were the last two: "NAGT Outcome 8: Reflect on personal strengths and challenges (e.g., in study design, safety, time management, independent and collaborative work)" and "NAGT Outcome 9: Demonstrate behaviors expected of professional geoscientists (e.g., time management, work preparation, collegiality, health and safety, ethics)." Students reflected on personal strengths and challenges (NAGT Outcome 8) and discussed time-management strategies in an informal way in their daily Slack discussion posts. Students wrote:

*"I also struggled with the excel worksheet today. It started making more sense towards the end, I will definitely have to go back and rewatch the meetings to grasp everything that is going on. For the GNSS sketch assignment, I'm not exactly sure what exaclty this questions is asking if anyone could help, thank you!"*

*"Today's work was not as confusing as the past few days. Having background knowledge on ArcMap definitely helped, but Cloud Compare took a while to maneuver. Just trying to keep up with the assignments and get the readings done. I'm trying to make it out on Sunday, though! I think the in-person field component will be really cool, and seeing other human beings would be awesome haha. As [student name] mentioned, interpreting the models can be tricky and applying them back to what we've been learning takes time, but really helps! Those connections do a great job to solidify the lessons."*

*"I think my biggest challenge today is interpreting all the models (DEM, hillshade, slope, etc) and what each one can be used for. I used USGS satellite images and classified them years ago in ArcMap for a project but I feel like I remember almost nothing from that so I'm a little lost!"*

*I'm still catching up from yesterday as well, but I feel significantly better than I did 24 hours ago! I remember just enough about ArcMap for it to be fun to figure out new challenges rather than frustrating, and I think that that was a nice boost after previous frustrations.*

### 5.2.2 Demographic outcomes

The cancellation of many field courses and change to remote instruction culminated in a more diverse course than UNC Earth Science majors typical demographic makeup. Students came from a wider variety of geographic regions, including six US states, one US territory, and one international location. Twenty-four percent of students (out of class of 23) were from

historically marginalized groups (American Indian or Alaska Native, Asian, Black or African American, Hispanic or LatinX,
and Multiracial) and 56 % were female compared to the 2011-2020 UNC Earth Science majors' averages of 17 % and 39 %,
respectively. Remote instruction may therefore aid in increasing representation in marginalized groups. At least 40 % of
students needed the course to meet degree requirements and most of the seven graduate students needed the expertise for their
graduate research.
**6 Lessons Learned and Implementation Recommendations**
Despite the challenging conditions under which this course was implemented, the course was highly successful overall by a
number of metrics, including frequently exemplary-level accomplishment on assessments and nearly all students passing the
course. When course-specific learning outcomes are considered, the vast majority were met, as indicated by assignment-
specific outcomes (section 5.1) as well as by their self reflection from the Slack daily discussion. In particular, students were
able to achieve Course-specific Outcomes A, C, D, and E (Section 2.3; Table 1) particularly well. Learning Outcome A (make
necessary calculations to determine the optimal survey parameters and survey design based on site conditions and available
time) was well-realised in terms of students' ability to understand the time it takes for post-processing and interpreting data of
a variety of types, and how one might improve upon the workflow. However, students did not receive the hands-on experience
they would have in the field. For example, they are not able to evaluate the time to set-up an RTK GNSS system, lay out
ground control points, survey them in and fly a UAS over the area with an appropriate team. This allows one to know the
spatial extent one can realistically cover in a given time. Students did learn the time it takes to post-process the imagery into
an SfM model as well as derivative products (e.g., rasters). Students also did not accomplish a sense of the time required to
conduct a TLS survey.  We realised in retrospect that Course Outcome B (Integrate GNSS targets with ground-based lidar and
SfM workflows to conduct a geodetic survey) was not fully accomplishable in the remote setting. Students were able to propose
and evaluate the design for ground control points in an SfM survey, but they were not able to actually "conduct" the survey.
Nor was the course able to provide an opportunity for similar experience in a lidar survey. If the course is taught remotely in
the future, this outcome should be rewritten to something more along the lines of "Recommend locations for a set of ground
control points for an SfM and/or TLS survey and critique surveys designed by others." The current Outcome B would be
appropriate for an in-person field course in its presented form.

519           The lowest level of accomplishment in the course came during the Day 8-9 assignments (5.1.4). As described, this is

likely because of a combination of difficulty and burnout. This course was moved to virtual because of safety concerns
surrounding COVID-19. However, the time commitment was kept the same as originally scheduled for in-person. As such, the
course was about two weeks (for three credits) fulltime (all day plus homework), similar to what would be expected for a
traditional in-person field-camp style course. This schedule proved exhausting with the online (Zoom lecture and office hours)
commitments for the course (morning and afternoon) combined with the computer-intensive nature of the assignments. In
particular, challenges in this format included 1) computational access (e.g., a good enough computer) and 2) access to the time

and space needed to complete the course. Several students dropped the course when they realised these constraints because of work and family obligations. However, of the 23 students who stayed enrolled in the course, 48 % received an A, 17 % a B, and 26 % a C, with A, B, or C marks comprising 91 % of the course. This demonstrates a high level of competence and performance for the vast majority of students. One student earned a D (corresponding to 60.0 % - 69.9 %) by completing 70 % of the assignments. This student expressed difficulty focusing for the length of time required for the course's pace. There was one student who earned an F which reflected participating and turning in only one day's worth of assignments. These students, while the minority (2 out of 23), should not be ignored. Studies suggest COVID exacerbated the ongoing mental health crisis among college students, increasing depression and anxiety (Son et al., 2020; Wang et al., 2020). The combination of COVID-19-related stress, virtual nature of the course, and intensity of workload likely contributed to feelings of anxiety in this course. We recommend, if this course is taught virtually in the future, to implement it as a longer interim session (minimum four weeks), quarter-, or semester-long course. Additionally, having computers available in a lab or on loan with the appropriate computational and software needs would be helpful. Students wrote in Slack reflections:

*"The only struggle I am having is my computing capabilities and it always crashing."*

*"I had to keep my computer running last night to generate the dense point cloud, but am glad to see that this*

*morning it has finally finished so that I can finish up the assignment.*

*"To improve the workflow when using this method in the future, a better computing device that can handle large*

*files would be better"*

If implemented as an intensive workshop, we recommend using at most four days' worth of material as presented here (eg., most of Unit 1). Any individual activity could be adapted as an assignment in an upper-division geomorphology or quantitative geoscience methods course. We are fairly certain that increasing the available time and support to complete the later assignments would mitigate the majority of the problem with lower student success, but we also suggest re-evaluating the later assignments for instructional clarity and supporting resources.

Lastly, student feedback and requests for additional offerings of the course indicate student appreciation of the course. One student wrote the instructor, "*I just wanted to thank you for the class. I have had an incredible journey during my university experience. Without this class being offered I truly do not know what I would have done. This has been a very trying time in my life and completing this course was the push I needed to continue through. I can't thank you enough for doing this. Not only offering the class but how flexible you were and understanding. Hands down one of the best professors I have had to date. You are an incredible teacher and I am very grateful that I took this class with you. Once again, from the bottom of my heart, thank you!*"

**Code/Data availability**
Not applicable
**Data availability**
Not applicable
**Competing interests**
The authors have no competing interests
**Author contributions**
Sharon Bywater-Reyes and Beth Pratt-Sitaula both contributed to field dataset collection used in the course. Both authors
contributed to the development of the curriculum presented. Both authors substantially wrote sections of the manuscript and
contributed to the revision process. Bywater-Reyes compiled the student evidence presented.
**Acknowledgments**
We thank Keith Williams and Erika Schreiber (UNAVCO), Ara Metz, Chelsie Romulo, and James Doerner for field data
collection support and the City of Greeley and the Poudre Learning Center for field site access. Special thanks to Melissa
Weinrich for insightful review and recommendations for revisions on this manuscript.

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

**Table 1. Activities by day and alignment with course-specific and NAGT learning outcomes.**

| Activity | Course-specific learning outcomes | NAGT capstone field learning outcomes |
|---|---|---|
| **Course Unit 1: SfM and GPS/GNSS**<br>Day 1 - Getting started with Structure from Motion (SfM) photogrammetry (https://serc.carleton.edu/NAGTWorkshops/online_field/activities/238996.html) | A. Survey design<br><br>C. Point cloud and DEM | 1, 2, 7 |
| Day 2a - GPS/GNSS Fundamentals (https://serc.carleton.edu/getsi/teaching_materials/high-precision/unit1.html) | A. Survey design<br><br>B. GNSS and geodetic survey<br><br>E. Justify tools/techniques | 1 |
| Day 2b - Post-processing GPS/GNSS Base Station Position (https://serc.carleton.edu/NAGTWorkshops/online_field/activities/239147.html) | B. GNSS and geodetic survey | 1 |
| Day 3a - Ground Control Points for Structure from Motion Activity (https://serc.carleton.edu/NAGTWorkshops/online_field/activities/239349.html) | A. Survey design<br><br>B. GNSS and geodetic survey | 1-5, 7, 9 |
| Day 3b - Structure from Motion for Analysis of River Characteristics Activity (https://serc.carleton.edu/NAGTWorkshops/online_field/activities/239350.html) | C. Point cloud and DEM<br><br>D. Geomorphic analysis | 1-5 |
| Day 4 - Working with Point Clouds in CloudCompare and Classifying with CANUPO (https://serc.carleton.edu/NAGTWorkshops/online_field/activities/240357.html) | C. Point cloud and DEM<br><br>D. Geomorphic analysis<br><br>E. Justify tools and techniques | 3-5 |
| Day 5 - SfM Feasibility Report assignment (https://d32ogoqmya1dw8.cloudfront.net/files/NAGTWorkshops/online_field/courses/sfm_feasibility_report.v2.docx) | A. Survey design<br><br>B. GNSS and geodetic survey<br><br>C. Point cloud and DEM<br><br>D. Geomorphic analysis<br><br>E. Justify tools/techniques | 3-6 |
| Day 6 - Optional field day | B. GNSS and geodetic survey | 1, 7, 9 |

| | | |
|---|---|---|
| **Course Unit 2: TLS, Topographic Differencing, and Method Comparison**<br>Day 7 - Introduction to Terrestrial Laser Scanning (TLS)<br>(https://serc.carleton.edu/NAGTWorkshops/online_field/activities/241028.html) | C. Point cloud and DEM<br><br>(E. Justify tools/techniques) | 3-7, 9 |
| Day 8a - Point Cloud and Raster Change Detection<br>(https://serc.carleton.edu/NAGTWorkshops/online_field/activities/241083.html) | C. Point cloud and DEM<br>E. Justify tools/techniques | 3-7, 9 |
| Day 8b - DEM of Difference<br>(https://serc.carleton.edu/NAGTWorkshops/online_field/activities/241138.html) | C. Point cloud and DEM<br>D. Geomorphic analysis | 3-7, 9 |
| Day 9 - OpenTopography Data Sources and Topographic Differencing<br>(https://serc.carleton.edu/NAGTWorkshops/online_field/activities/241410.html) | C. Point cloud and DEM<br>D. Geomorphic analysis | 3-6, 9 |
| Day 10 - Methods Comparison Report<br>(https://d32ogoqmya1dw8.cloudfront.net/files/NAGTWorkshops/online_field/courses/methods_comparison_report.docx) | A. Survey design<br>C. Point cloud and DEM<br>D. Geomorphic analysis<br>E. Justify tools/techniques | 3-6 |
| Day 11 - Presentations | D. Geomorphic analysis<br>E. Justify tools/techniques | 6-7, 9 |




**Table 2 - Example rubric showing percentage scoring used to assess course activities.**

| | Exemplary (75-100% points) | Basic (50-75% points) | Minimal effort (25-50%) | Nonperformance (0-25%) |
|---|---|---|---|---|
| General Considerations | Exemplary work will not just answer all components of the given question but also answer correctly, completely, and thoughtfully. Attention to detail, as well as answers that are logical and make sense, is an important piece of this. | Basic work may answer all components of the given question, but answers are incorrect, ill-considered, or difficult to interpret given the context of the question. Basic work may also be missing components of a given question. | Minimal performance occurs when student answers simply do not make sense and are incorrect. | Nonperformance occurs when students are missing large portions of the assignment. |



**Figures**

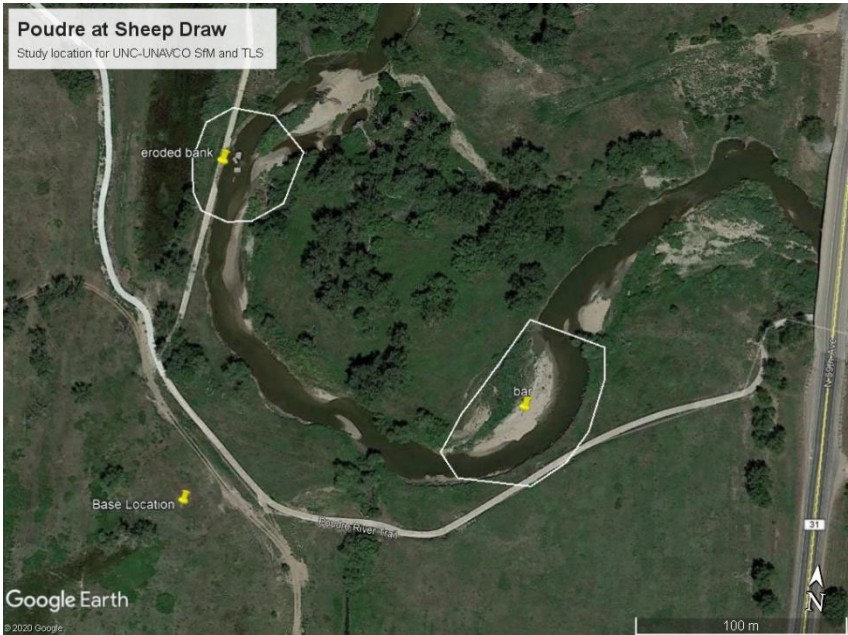


**Figure 1. Inset: Map (© Google Earth) of the Cache la Poudre River Watershed, located in northern Colorado, US. The study site**
**at Sheep Draw has two areas of interest, Area of Interest 1 on an eroded bank and Area of Interest 2, a cutbank and point bar.**

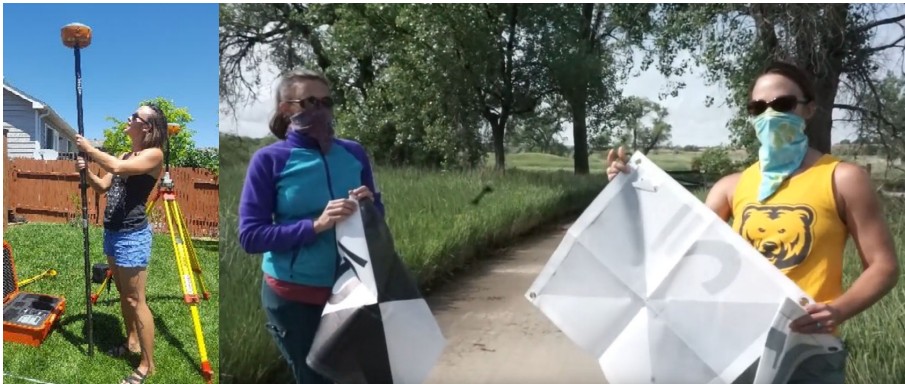


**Figure 2. Base and Kinematic GNSS methods (left) and example of ground control (right) surveyed for use in GNSS and SfM**
**activities.**



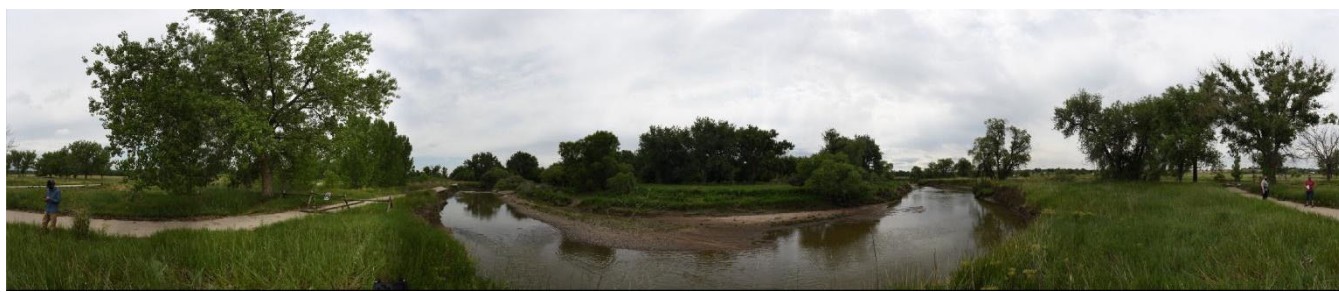



**Figure 3. The top shows the Terrestrial Laser Scanner (TLS) photograph from a scan location whereas the bottom shows the**
**associated point cloud at the Cache la Poudre River site. Courtesy UNAVCO.**



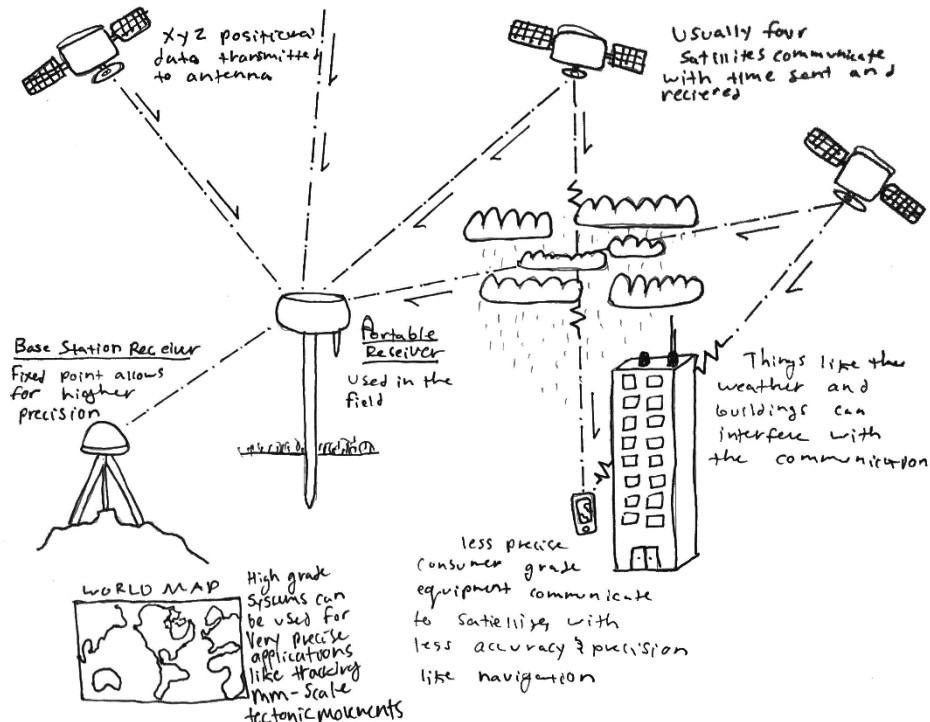

- Xyz positional data transmitted to antenna
- Usually four satellites communicate with time sent and recieved
- Base Station Reciever — Fixed point allows for higher precision
- Portable Reciever — Used in the field
- Things like the weather and buildings can interfere with the communication
- WORLD MAP — High grade systems can be used for very precise applications like tracking mm-scale tectonic movements
- less precise consumer grade equipment communicate to satellites with less accuracy & precision like navigation



**Figure 4: Student sketch of how GNSS works, including disruptions and applications thereof demonstrating theoretical**
**understanding of GNSS (created by student in course for Day 2 activity; student not disclosed to comply with Institutional Review**
**Board).**

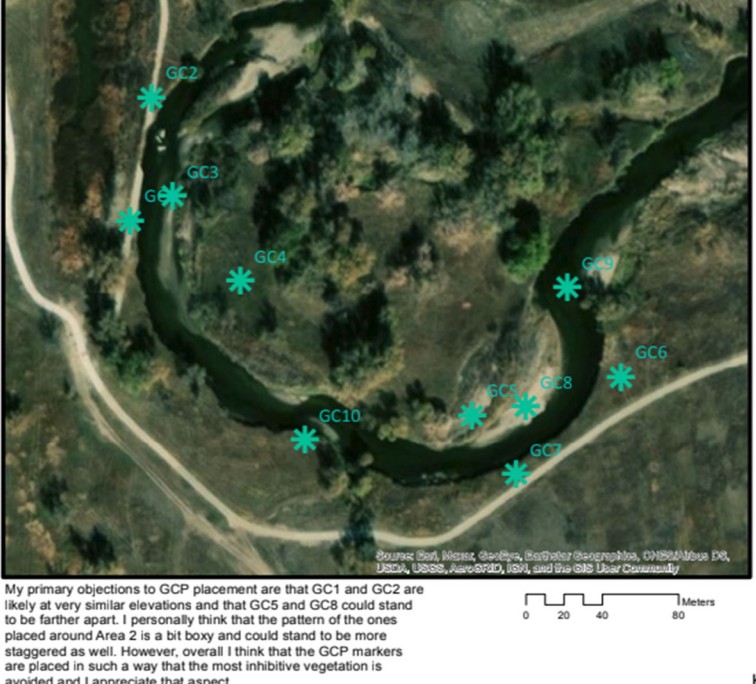


**Figure 5. Student map of ground control points (GCPs) used in SfM activity (created by student in course; student not disclosed to**
**comply with Institutional Review Board). Through a group discussion on Day 2, students discussed whether GCPs were adequately**
**placed and suggested implementation improvements. Imagery source: ArcGIS® software by Esri.**

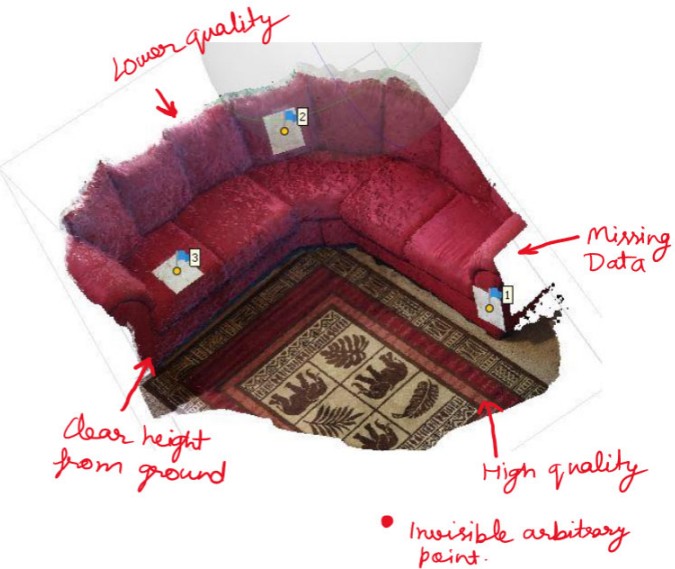


**Figure 6: Student SfM product from Day 1 exercise (created by student in course; student not disclosed to comply with Institutional**
**Review Board). Student successfully assessed relative data quality as indicated by student's markup, and where data was missing**
**or of low quality.**




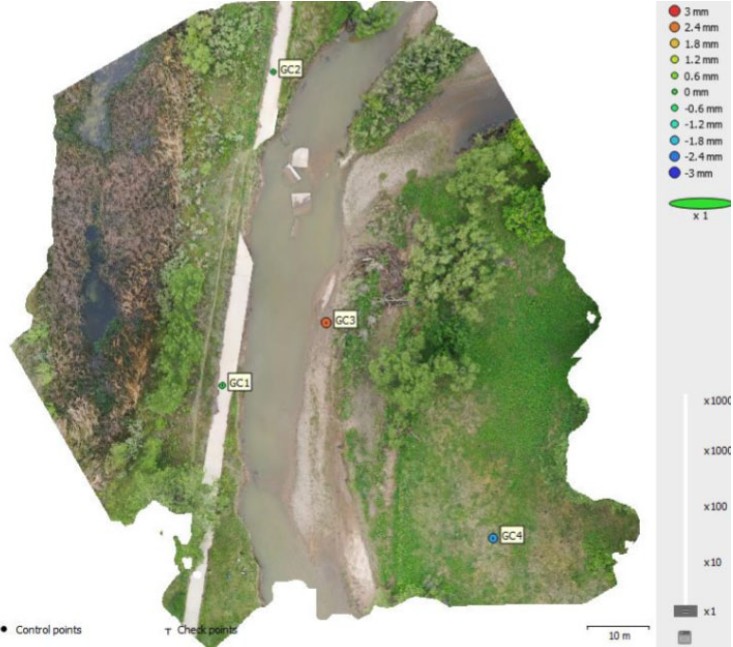

**Figure 7. Student-generated colored SfM point cloud of their area of interest showing GCP error ellipsoids used by the student in**
**their SfM error analysis (created by student in course; student not disclosed to comply with Institutional Review Board).**