# Peer review of "A Remote Field Course Implementing High-Resolution Topography Acquisition and Applications Applied to Geomorphology"

_Geoscience Communication, 2021_

## Author Comment (AC3)

*We want to thank Reviewer 1 for the detailed and thoughtful review, which really helped us plan revisions to improve the manuscript.*

Reviewer 1

Thank you for the opportunity to read this submission. It is clear that the authors have put a lot of time and effort into creating a remote field experience for their students. Unfortunately, this is overshadowed by the confusing set up for the paper and lack of evidence presented to substantiate their claims. I believe that major restructuring is needed for this to be publishable. Below, I describe the bigger picture strengths and challenges that I observed in each section. This is followed by more minor (line-by-line) comments.

*We concur that restructuring is needed and details of the plans are given in specific locations below.*

**Title:**

Though the title describes the field course adequately, I find the duplication of almost the exact same word ("applications applied") to be hard to follow.

*We will remove "Applied" and leave the title as "A Remote Field Course Implementing High-Resolution Topography Acquisition with Geomorphologic Applications"*

**Abstract:**

The abstract is well written, but it is too focused on the course context (what is essentially background information). This takes away from describing the impact of the curriculum that you have developed. You only say that "students met the majority of the NAGT field capstone learning outcomes" which is rather vague. I want to know which outcomes specifically were met (or a selection of them at least) and how you measured this. I recommend reducing the details of the course context to 2-3 sentences and focusing more on the project outcomes and recommendations.

*Abstract will be rewritten based on this guidance and to match the final version of the paper which reflects these issues. We will remove some details about the course and add details about outcomes, as outlined below.*

**Introduction:**

This section is largely course context information that would be much more appropriate after the relevant literature context is presented. Furthermore, the level of detail about the course context is excessive for an introduction and should instead be a separate section entirely.

The introduction does not flow logically. It begins by describing the course, then talks about the remote field data collection and activities (and where they are available), then goes back to the course context and the data sources, which is then followed by more course context (demographic information). A more common approach would be to discuss the course context, then discuss the specific activity and how it was developed, then follow with where the completed activity may be found. However, as indicated above, I believe this would be more effective as a separate section that follows the introduction.

*We are planning to restructure the paper to be:*

The last paragraph of Section 1.0 seems to be suggesting that COVID-19 has caused this particular course to be more diverse. This claim is unsubstantiated and should be removed. There is no comparison to previous year's demographic data, and even that would not be enough to support a causal link. I believe that the literature cited regarding barriers to fieldwork is incredibly important, but it seems out of place here. Perhaps have a subsection of the introduction that is dedicated to synthesizing relevant literature on the barriers to fieldwork and the potential for remote field courses to help address some of those barriers?

> *We will move demographic data to "Course implementation and outcomes" section (as a result) and will add a comparison with UNC's majors' demographics which support this claim. We will also use different language when referring to groups historically marginalized in the geosciences.*

Section 1.1 doesn't seem to be connected to prior or subsequent sections and feels disjointed. I think that moving course context information to a separate section will help, and then this section may be framed as supporting literature context for the work.

Section 1.2 (Overview of Modules and Learning Objectives) is repetitive of prior and subsequent information. I don't think there is any need to give an overview before presenting day-by-day details that are not overly long. I suggest creating two tables to help streamline this information: 1) A table that compares the course learning outcomes to the NAGT field outcomes, and 2) A table that provides an overview of the itinerary (the information currently presented in L118-160).

> *We agree and will be restructuring as described above.*
>
> *We plan to use several different tables similar to what the review suggests - in the new sections 2) and 4) in order to more clearly present the alignment between a given course activity, activity learning outcomes, NAGT learning outcomes, supporting student data/evidence, and explanations for why evidence supports claims made. We will plan to use tables and more concise text sections than the original manuscript to accomplish this.*

**Curriculum description (Section 2):**

This section includes student artefacts and performance data. I think it would be helpful to state this in an overarching title or introductory paragraph for the section, so that readers are expecting it. Something like "unit descriptions and evidence of student success"? Evidencing your curriculum's effectiveness is incredibly important, and this is buried at present.

On a related note, I do not see any mention of human ethics (i.e., Institutional Review Board) approval. The figure captions reference compliance, but it is still not clear if the study went through a review process. If human subjects approval was confirmed through the submission process, please disregard my comment.

*We will add a short methods section describing what data we have and that we have IRB approval to use course artifacts. We will also explain that because this course was not a designed educational research study (but as a response to COVID19), there are inherent limitations. We will also describe how student work was assessed and provide an example rubric. This will provide context for the percentage scores currently presented.*

Some of this section repeats data sources / curricula that were adapted, both of which were described in detail much earlier in the manuscript. I think you could remove that specific information from the introduction in favour of only including it here, as it is much more meaningful when placed in context.

*Paper restructuring described above should address this issue as well.*

**Lessons learned (Section 3):**

I appreciate the authors' reflections here, but they would be strengthened if they were supported by evidence (from the literature and/or data in the study).

Student feedback is mentioned in this section without any description of how these data were collected or how the claim of "student appreciation of the course" is supported.

*We will be more transparent that a few students provided unsolicited appreciation for the course. We will include statements from students' daily journal entries where relevant. We will pull from the "Course implementation and outcomes" to support assertions made in the "lessons learned and recommendations" section.*

**Technical (line-by-line) comments:**

- L10: I found this phrasing a bit awkward. Perhaps start with a word other than "with"?

- L14: The use of "mock" and "real" undermines the virtual field aspect. I suggest "remote" and "authentic".

- L15-18: Suggest moving the objectives and course content earlier in the paragraph; e.g., after the first sentence. This way things flow a bit more logically and you are presenting the course context before describing a specific activity within it. (However, see my earlier comments about the abstract more generally - I think you could cut this background information down substantially)

*Edits will be made as suggested*

- L24: Elaborate on what is meant by "field tradition".

- L26: Evidence needed to support the claim that "the majority" of courses were redesigned for remote delivery.

*We will cite records from*

*https://nagt.org/nagt/teaching_resources/field/summer_2020_virtual_field_camp.html*

*that show a record of virtual field offerings; many field courses were canceled and therefore this resource was compiled, which is how external students found the course.*

*https://nagt.org/nagt/publications/trenches/v11-n1/online_field_experiences.html*

- L38-40: Calling this crowd-sourcing curriculum development undersells the work that went into the project and the evidence that informed it. There were several facilitated, structured working groups dedicated to different topics and a concerted effort to articulate consensus learning outcomes. I appreciate that you are trying to be brief, but I think that more care is needed here. If the authors were not involved in the working groups, it is sufficient to say that the curriculum was added to this resource collection.

*We will remove this language and use "working groups" and "community developed" instead.*

- L49: What is "workshop style"? Please include more pedagogical detail or a citation here.

*We will remove "workshop style" and provide details on session formats and duration, including that each day was composed of multiple synchronous working sessions with asynchronous work time in between.*

- L50: How many is "several"? How does this compare to overall course numbers? Even just a rough idea would be helpful here.

*Will add actual number of students*

- L67: I don't think "incurred" is the right word here. Also, much more detail and relevant citations are needed to explain the affective factors that are being referred to.

*Will change "incurred" to "experience" if we keep this sentence*

- L68: Change "major's" to "major". Also, what is meant by "most diverse"?

*As mentioned above, we will move this to outcomes section, add context, and expand upon what is meant based on the comparison that can be made and change the language as described in response to L69 comment*

- L68-70: Remove spaces between numbers and percentage signs.

*We will follow the journal's style guideline*

- L69: The term "underrepresented minorities" is deficit-framed and puts the burden on students rather than recognizing the systems that have excluded them from STEM (see this blog post for one example of a critique: https://cacm.acm.org/blogs/blog-cacm/245710-underrepresented-minority-considered-harmful-racist-language/fulltext). Please consider if there is a more appropriate term that may be used for your context.

*Thank you for pointing this out. If the reviewer has a specific term they recommend, we would be happy to use that. Reviewing current use by NSF, Geopaths, and others it seems that there is not yet consensus on the best terminology. We propose to use "historically marginalized groups" and list them.*

- L79: Quotation marks should end before the citation. Also, a page number should be included with the quotation.

> *We will fix as suggested.*

- L110: This seem to be different than the course learning outcomes stated in the abstract and earlier on in the introduction. Or are these field activity specific learning outcomes?

> *In the new sections 2) and 4) we will more clearly and consistently specify which activity, course, and NAGT learning outcomes that are supported by particular pieces of student evidence.*

- L189-190: You mention GPS and GNSS much earlier in the paper, but this is the first time they are defined.

> *We will make sure all acronyms are defined the first time they are used*

- L238-240: Quote all learning outcomes here rather than describing vaguely. Also, you don't present evidence here that students met these outcomes, so it is inaccurate to say "this allowed them to meet the student learning outcome". Rather, the activities are designed to address or align with specific outcomes.

> *We will be clearer about this as suggested and only make assertions that are backed up by evidence. We will be adding some additional student evidence than is in the original manuscript. We have consulted with an educational researcher to attain additional recommendations on what evidence to use and how to present it.*

- L257: Same comment as above re: "allowed students to meet".

> *See response to L238*

- L322: No evidence is provided to suggest that students "enhanced their skills".

> *See response to L238*

- L327: Change "student" to "students".

> *Will change*

- L333-335: Elaborate on why this is a significant outcome.

> *Will elaborate on this illustrating a student's ability to apply the breadth of methods used in the class in a unique application and highlight this as evidence of the student's self-efficacy and interest in the topic. May be combined with evidence from one or more other students.*

- L343-345: I believe this, but it would be better supported with data from the students. Or are you saying that it was exhausting for the instructors? Or both?

> *Both; will clarify and back up with student evidence from students' discussion posts*

- L363-364: Do you have any suggestions for how these may be better addressed?

*We do not immediately have a solution but will continue to ponder it during the rewriting process.*

- L364-365: Evidence is needed to substantiate this claim.

*We agree and will back with evidence*

- L398: Atchison is listed twice here.

*Will fix*

- Figure 4: Are these students in the images? If so, did they consent to their images being included?

*Yes; covered by IRB; we will add this to the caption*

---

## Author Comment (AC4)

*We thank Reviewer 2 for their constructive and thought-provoking comments.*

Reviewer 2

Bywater-Reyes and Prat-Sitalula present the structure of a remote field course that focuses on the application of remote sensing to geomorphology. The value of the course topic is clearly articulated and well cited. The course leveraged several 3D datasets acquired through different remote-sensing techniques. The students were given tasks of acquiring, building, manipulating, and analyzing data, and were evaluated on task completion (e.g., data measurements), report writing, hypothesis testing, interpretations, and data quality assessment. The authors describe how the course meets learning outcomes determined by the NAGT Teaching with Online Field Experiences.

Modern remote-sensing methods and data introduced in this field course have applications to both research and industry, so a description of how this course was implemented remotely has value to the geoscience education community. Some major strengths of this manuscript are 1) examples of the students' work which provide readers with a clear picture of student deliverables; 2) links to the published teaching materials used within the course; and 3) a schedule that depicts a logical flow of topics that culminates with a larger-scale project. Should someone want to implement a similar course, this document serves as a good resource.

As a *research* article, the manuscript would benefit from more data on student assessment and/or engagement. Some thoughts on how to enhance the research contributions of this manuscript are as follows:

*We agree that better data on student assessment is needed. We plan to restructure the manuscript fairly significantly as follows (based also on feedback from another reviewer).*

1. *Introduction*
    1.1. *Background (on why/what we did)*
    1.2. *Value of course content geologically*
    1.3. *Value of remote field learning to remove barriers to participation*
2. *Overview of the course*
3. *Methods in this study*
4. *Course implementation and outcomes*
5. *Lessons learned and implementation recommendations*

*In the new sections 2) and 4) we will also more clearly and consistently specify which activity, course, and NAGT learning outcomes that are supported by particular pieces of student evidence.*

- A comparison of student performances in this course to student performances in another course on remote sensing would add research value to the manuscript. Was a field-based version of this course previously offered at the University of Northern CO, or is there literature that documents a field-based remote-sensing-geomorph course with student assessments? It's unclear whether this course on remote-sensing data and geomorphology

is novel in the university's course offerings, or if just the remote format in which it was taught is novel.

*Unfortunately, this was the only implementation of this particular course and although aspects of this course are found in other UNCO courses, it is sufficiently different and during the general upheaval from COVID, that direct comparisons do not seem valid. The curriculum in its current form was developed specifically because of the need to teach field methods remotely. We will make this clearer. The published curriculum combined with this manuscript provide guidance on implementation and outcomes.*

- Student engagement is referred to with regards to the difficulties surrounding computer access, long hours on Zoom, and a quote at the end of the paper by a single student. More student accounts of the remote-learning conditions and/or student reactions to the course would substantiate "difficult conditions" (line 356) and "student appreciation" (line 372) in the Lessons Learned section.

    *We will add evidence from student discussion forums that support these claims.*

- There are general descriptions of what deliverables and abilities of the students were evaluated, and mastery levels were provided as percentages. The authors mention that as-needed problem solving and decision making were two of the NAGT learning outcomes that may have not been met. Is there an area of the projects or deliverables in which student performances were lower in general, and does this reveal a shortcoming in the remote nature of how the course was taught? I would be interested in this!

    *Good point! We will more clearly state in the "Lessons learned and implementation recommendations" section the skills that students were strongest and weakest on during this course (with evidence drawn from student data).*

**Minor edits and suggestions:**
Formatting: Structure from Motion photogrammetry (SfM) should be written first with "photogrammetry" attached, and subsequently referred to as SfM without the need to redefine the acronym in subsequent sections. Similarly, TLS only needs to be written out once if the authors intend to use the acronym TLS throughout the remainder of the manuscript.

*We will change as recommended*

Line 68: Diversity is quantified for the student population for this course. The authors say that it is the "most diverse major's course by the instructor." To substantiate this, how was this diversity assessed, and how does it compare quantitatively to previous courses taught by the instructor?

*We will more clearly define this and compare to demographics of other similar courses.*

Line 146 and 147: "lidar" is inconsistent with the use of the form "LiDAR" in the rest of the manuscript.

*We will make it consistent throughout the paper, following the style convention of Geosci Comm. If GC does not have a convention, we will use "lidar".*

Line 180: How did students access Agisoft Metashape? Did they use a remote connection to on-site computers, use the trial version, or were they provided with licenses from the school or another party?

*They were provided with a trial version which the instructor requested from Agisoft*

Tenses change several times throughout the paper, e.g., the sentence starting in 232 vs the sentence in 234, and sentence 218 vs 221.

*We will proofread for consistency on tense.*

Line 352: Depending on how the mention of diversity is handled in line 68, the mention of "diverse" with regards to students may be omitted.

*We will make sure that all references to "diversity" within the paper are self-consistent.*

---

## Author Response (AR1)

*We want to thank Reviewer 1 for the detailed and thoughtful review, which really helped us plan revisions to improve the manuscript.*

Reviewer 1

Thank you for the opportunity to read this submission. It is clear that the authors have put a lot of time and effort into creating a remote field experience for their students. Unfortunately, this is overshadowed by the confusing set up for the paper and lack of evidence presented to substantiate their claims. I believe that major restructuring is needed for this to be publishable. Below, I describe the bigger picture strengths and challenges that I observed in each section. This is followed by more minor (line-by-line) comments.

*We have substantially rewritten the introduction and added many sections to the manuscript, including a methods, results (with lots of new evidence), discussion, and recommendations section and reframed and edited the previous section explaining course implementation with very limited assessment information as a course overview section.*

**Title:**

Though the title describes the field course adequately, I find the duplication of almost the exact same word ("applications applied") to be hard to follow.

*We changed to "A Remote Field Course Implementing High-Resolution Topography Acquisition with Geomorphic Applications"*

**Abstract:**

The abstract is well written, but it is too focused on the course context (what is essentially background information). This takes away from describing the impact of the curriculum that you have developed. You only say that "students met the majority of the NAGT field capstone learning outcomes" which is rather vague. I want to know which outcomes specifically were met (or a selection of them at least) and how you measured this. I recommend reducing the details of the course context to 2-3 sentences and focusing more on the project outcomes and recommendations.

*Abstract rewritten to match revisions of the paper which reflects these issues. We removed some details about the course and added details about outcomes.*

**Introduction:**

This section is largely course context information that would be much more appropriate after the relevant literature context is presented. Furthermore, the level of detail about the course context is excessive for an introduction and should instead be a separate section entirely.

The introduction does not flow logically. It begins by describing the course, then talks about the remote field data collection and activities (and where they are available), then goes back to the course context and the data sources, which is then followed by more course context (demographic information). A more common approach would be to discuss the course context, then discuss the specific activity and how it was developed, then follow with where the completed activity may be found. However, as indicated above, I believe this would be more effective as a separate section that follows the introduction.

*We substantially revised the organization according to the following:*

*1 Introduction (revised)*

*2 Course overview and learning outcomes (revised)*

*3 Methods (new)*

*4 Course Implementation and Assessment Approach (substantially revised)*

*5 Results (new)*

*6 Lessons Learned and Implementation Recommendations (new)*

The last paragraph of Section 1.0 seems to be suggesting that COVID-19 has caused this particular course to be more diverse. This claim is unsubstantiated and should be removed. There is no comparison to previous year's demographic data, and even that would not be enough to support a causal link. I believe that the literature cited regarding barriers to fieldwork is incredibly important, but it seems out of place here. Perhaps have a subsection of the introduction that is dedicated to synthesizing relevant literature on the barriers to fieldwork and the potential for remote field courses to help address some of those barriers?

> *We moved demographic data to Section 5.2 and removed any additional claims. We added information from the literature about barriers to traditional field courses and benefits to alternatives (in Section 1.3). We changed language to "historically marginalized" and defined.*

Section 1.1 doesn't seem to be connected to prior or subsequent sections and feels disjointed. I think that moving course context information to a separate section will help, and then this section may be framed as supporting literature context for the work.

Section 1.2 (Overview of Modules and Learning Objectives) is repetitive of prior and subsequent information. I don't think there is any need to give an overview before presenting day-by-day details that are not overly long. I suggest creating two tables to help streamline this information: 1) A table that compares the course learning outcomes to the NAGT field outcomes, and 2) A table that provides an overview of the itinerary (the information currently presented in L118-160).

> *We changed section 1 to be an introduction including sections on background on context of course, value of course topic, and value of remote learning. Information from Section 1.2 has been moved to section 2 and a table of course-specific and NAGT outcomes with alignment between activities and objectives indicated (Table 1).*

**Curriculum description (Section 2):**

This section includes student artefacts and performance data. I think it would be helpful to state this in an overarching title or introductory paragraph for the section, so that readers are expecting it. Something like "unit descriptions and evidence of student success"? Evidencing your curriculum's effectiveness is incredibly important, and this is buried at present.

On a related note, I do not see any mention of human ethics (i.e., Institutional Review Board) approval. The figure captions reference compliance, but it is still not clear if the study went through a review process. If human subjects approval was confirmed through the submission process, please disregard my comment.

*We added a methods section that describes the data we had available and also added IRB approval information. We explained that because this course was not a designed educational research study (but as a response to COVID19), there are inherent limitations.*

*We added a section (Section 4 Course Implementation and Assessment Approach) that describes how student work was assessed and provided an example rubric (Table 2). This provides context for the percentage scores presented.*

Some of this section repeats data sources / curricula that were adapted, both of which were described in detail much earlier in the manuscript. I think you could remove that specific information from the introduction in favour of only including it here, as it is much more meaningful when placed in context.

*We separated course implementation (section 4) from outcomes (section 5). Section 5 has substantial new analysis and presentation of artifacts not included in the original manuscript, organized by learning outcome with evidence supporting whether the outcome was met or not, and to what extent.*

**Lessons learned (Section 3):**

I appreciate the authors' reflections here, but they would be strengthened if they were supported by evidence (from the literature and/or data in the study).

Student feedback is mentioned in this section without any description of how these data were collected or how the claim of "student appreciation of the course" is supported.

*We will be more transparent that a few students provided unsolicited appreciation for the course. We will include statements from students' daily journal entries where relevant. We will pull from the "Course implementation and outcomes" to support assertions made in the "lessons learned and recommendations" section.*

**Technical (line-by-line) comments:**

- L10: I found this phrasing a bit awkward. Perhaps start with a word other than "with"? *Edited as suggested.*

- L14: The use of "mock" and "real" undermines the virtual field aspect. I suggest "remote" and "authentic". *Edited as suggested.*

- L15-18: Suggest moving the objectives and course content earlier in the paragraph; e.g., after the first sentence. This way things flow a bit more logically and you are presenting the course context before describing a specific activity within it. (However, see my earlier comments about the abstract more generally - I think you could cut this background information down substantially)

*Edited as suggested.*

- L24: Elaborate on what is meant by "field tradition". *Removed entirely. No longer relevant.*

- L26: Evidence needed to support the claim that "the majority" of courses were redesigned for remote delivery.

> *To support this claim, we cited the following and were more explicit about additional evidence.*

> Egger, A. *et al.* Teaching with online field experiences: New resources by the community, for the community. *In The Trenches* **11**, (2021).

- L38-40: Calling this crowd-sourcing curriculum development undersells the work that went into the project and the evidence that informed it. There were several facilitated, structured working groups dedicated to different topics and a concerted effort to articulate consensus learning outcomes. I appreciate that you are trying to be brief, but I think that more care is needed here. If the authors were not involved in the working groups, it is sufficient to say that the curriculum was added to this resource collection.

> *We removed this language and changed to "working groups" and "community developed" instead and cited the project.*

- L49: What is "workshop style"? Please include more pedagogical detail or a citation here.

> *We defined "workshop style" and provide details on session formats and duration, including that each day was composed of multiple synchronous working sessions with asynchronous work time in between.*

- L50: How many is "several"? How does this compare to overall course numbers? Even just a rough idea would be helpful here.

> *Added actual number of students*

- L67: I don't think "incurred" is the right word here. Also, much more detail and relevant citations are needed to explain the affective factors that are being referred to.

> *Edited as suggested.*

- L68: Change "major's" to "major". Also, what is meant by "most diverse"?

> *Edited as suggested and explicitly explained what is meant (new Section 5.2.2)*

- L68-70: Remove spaces between numbers and percentage signs.

> *We will follow the journal's style guideline, which we believe has the space.*

- L69: The term "underrepresented minorities" is deficit-framed and puts the burden on students rather than recognizing the systems that have excluded them from STEM (see this blog post for one example of a critique: https://cacm.acm.org/blogs/blog-cacm/245710-underrepresented-minority-considered-harmful-racist-language/fulltext). Please consider if there is a more appropriate term that may be used for your context.

> *Thank you for pointing this out. Reviewing current use by NSF, Geopaths, and others it seems that there is not yet consensus on the best terminology. We changed to "historically marginalized groups" and listed them.*

- L79: Quotation marks should end before the citation. Also, a page number should be included with the quotation.

*Edited as suggested.*

- L110: This seem to be different than the course learning outcomes stated in the abstract and earlier on in the introduction. Or are these field activity specific learning outcomes?

*Defined Course- and NAGT outcomes and explicitly provided evidence for each in new organization*

- L189-190: You mention GPS and GNSS much earlier in the paper, but this is the first time they are defined.

*Defined all acronyms first time they are used*

- L238-240: Quote all learning outcomes here rather than describing vaguely. Also, you don't present evidence here that students met these outcomes, so it is inaccurate to say "this allowed them to meet the student learning outcome". Rather, the activities are designed to address or align with specific outcomes.

*Separation of course overview with outcomes listed from implementation (section 4) and evidence of outcomes (section 5) has remedied this*

- L257: Same comment as above re: "allowed students to meet".

*See response to L238*

- L322: No evidence is provided to suggest that students "enhanced their skills".

*See response to L238*

- L327: Change "student" to "students".

*Edited as suggested.*

- L333-335: Elaborate on why this is a significant outcome.

*Elaborated on this illustrating a student's ability to apply the breadth of methods used in the class in a unique application and highlight this as evidence of the student's self-efficacy and interest in the topic.*

- L343-345: I believe this, but it would be better supported with data from the students. Or are you saying that it was exhausting for the instructors? Or both?

*Both; clarified and backed up with student evidence from students' discussion posts*

- L363-364: Do you have any suggestions for how these may be better addressed?

*We added a section on NAGT outcomes and in doing so (along with accompanying new evidence) discovered this outcomes was met to some extent. We discussed this in section 5.2.1.*

- L364-365: Evidence is needed to substantiate this claim.

*Removed*

- L398: Atchison is listed twice here.

*Fixed*

- Figure 4: Are these students in the images? If so, did they consent to their images being included?

*Yes; covered by IRB; added info to the captions as well as in new methods section*

*We thank Reviewer 2 for their constructive and thought-provoking comments.*

Reviewer 2

Bywater-Reyes and Prat-Sitalula present the structure of a remote field course that focuses on the application of remote sensing to geomorphology. The value of the course topic is clearly articulated and well cited. The course leveraged several 3D datasets acquired through different remote-sensing techniques. The students were given tasks of acquiring, building, manipulating, and analyzing data, and were evaluated on task completion (e.g., data measurements), report writing, hypothesis testing, interpretations, and data quality assessment. The authors describe how the course meets learning outcomes determined by the NAGT Teaching with Online Field Experiences.

Modern remote-sensing methods and data introduced in this field course have applications to both research and industry, so a description of how this course was implemented remotely has value to the geoscience education community. Some major strengths of this manuscript are 1) examples of the students' work which provide readers with a clear picture of student deliverables; 2) links to the published teaching materials used within the course; and 3) a schedule that depicts a logical flow of topics that culminates with a larger-scale project. Should someone want to implement a similar course, this document serves as a good resource.

As a *research* article, the manuscript would benefit from more data on student assessment and/or engagement. Some thoughts on how to enhance the research contributions of this manuscript are as follows:

> *We substantially revised the organization according to the following:*
>
> *1 Introduction (revised)*
>
> *2 Course overview and learning outcomes (revised)*
>
> *3 Methods (new)*
>
> *4 Course Implementation and Assessment Approach (substantially revised)*
>
> *5 Results (new)*
>
> *6 Lessons Learned and Implementation Recommendations (new)*

A comparison of student performances in this course to student performances in another course on remote sensing would add research value to the manuscript. Was a field-based version of this course previously offered at the University of Northern CO, or is there literature that documents a field-based remote-sensing-geomorph course with student assessments? It's unclear whether this course on remote-sensing data and geomorphology is novel in the university's course offerings, or if just the remote format in which it was taught is novel.

> *Unfortunately, this was the only implementation of this particular course and although aspects of this course are found in other UNCO courses, it is sufficiently different and during the general upheaval from COVID, that direct comparisons*

*do not seem valid. The curriculum in its current form was developed specifically because of the need to teach field methods remotely. We will make this clearer. The published curriculum combined with this manuscript provide guidance on implementation and outcomes.*

- Student engagement is referred to with regards to the difficulties surrounding computer access, long hours on Zoom, and a quote at the end of the paper by a single student. More student accounts of the remote-learning conditions and/or student reactions to the course would substantiate "difficult conditions" (line 356) and "student appreciation" (line 372) in the Lessons Learned section.

  *We added evidence from student discussion forums that support these claims.*

- There are general descriptions of what deliverables and abilities of the students were evaluated, and mastery levels were provided as percentages. The authors mention that as-needed problem solving and decision making were two of the NAGT learning outcomes that may have not been met. Is there an area of the projects or deliverables in which student performances were lower in general, and does this reveal a shortcoming in the remote nature of how the course was taught? I would be interested in this!

  *Good point! We have an entire new section (5) dedicated to student evidence for coure- and NAGT outcomes. We more clearly state which were met and to what extent and discuss in "Lessons learned and implementation recommendations" section the skills that students were strongest and weakest on during this course (with evidence drawn from student data).*

**Minor edits and suggestions:**
Formatting: Structure from Motion photogrammetry (SfM) should be written first with "photogrammetry" attached, and subsequently referred to as SfM without the need to redefine the acronym in subsequent sections. Similarly, TLS only needs to be written out once if the authors intend to use the acronym TLS throughout the remainder of the manuscript.

  *Edited as suggested.*

Line 68: Diversity is quantified for the student population for this course. The authors say that it is the "most diverse major's course by the instructor." To substantiate this, how was this diversity assessed, and how does it compare quantitatively to previous courses taught by the instructor?

  *We explicitly define diversity and compare to typical demographics for our department's majors*

Line 146 and 147: "lidar" is inconsistent with the use of the form "LiDAR" in the rest of the manuscript.

  *Changed to lidar as suggested*

Line 180: How did students access Agisoft Metashape? Did they use a remote connection to on-site computers, use the trial version, or were they provided with licenses from the school or another party?

> *They were provided with a trial version which the instructor requested from Agisoft; this was added*

Tenses change several times throughout the paper, e.g., the sentence starting in 232 vs the sentence in 234, and sentence 218 vs 221.

> *Changed for consistency to past tense throughout*

Line 352: Depending on how the mention of diversity is handled in line 68, the mention of "diverse" with regards to students may be omitted.

> *Made "diversity"consistent within the paper are self-consistent.*

---

## Author Response (AR2)

Thanks so much for taking time to review this again.

line 236: add an "s" to "day", since you are referring to two days

Made change as recommended

line 311: remove "available" since it is redundant

Made change as recommended